# Gully Head-Cuts Inventory and Semi-Automatic Gully Extraction Using LiDAR and Topographic Openness—Case Study: Covurlui Plateau, Eastern Romania

**Ionut-Costel Codru** [1,2,*], **Lilian Niacsu** [2], **Andrei Enea** [2] **and Latifa Bou-imajjane** [3]

1 Research Center with Integrated Techniques for Atmospheric Aerosol Investigation in Romania, RECENT AIR, Laboratory of Interdisciplinary Research in Geo-Chemistry of Rural Areas, Environmental Quality Monitoring Station for Geographic Research, "Alexandru Ioan Cuza" University of Iasi, 707290 Iasi, Romania
2 Faculty of Geography & Geology, "Alexandru Ioan Cuza" University of Iasi, Carol I Blvd. 11, 700506 Iasi, Romania; lilian.niacsu@uaic.ro (L.N.); andrei.enea@uaic.ro (A.E.)
3 Geology Department, Faculty of Science, Ibn Zohr University, Agadir 80060, Morocco; latifa.bou-imajjane@edu.uiz.ac.ma
* Correspondence: ionut.codru@uaic.ro

**Abstract:** The Covurlui Plateau, a subunit of the Moldavian Plateau located in eastern Romania, possesses a high natural agricultural potential, significantly impacted by soil erosion, particularly gully erosion. The only inventory in the Moldavian Plateau that comprises approximately 9000 gullies extracted from topographical maps was conducted during the 90s. Nowadays, with the advent of advanced techniques and geodata, such as GIS software, aerial photograms, high-resolution satellite images, and high-resolution digital elevation models, we aim to achieve an (1) up-to-date comprehensive inventory of gully head-cuts and (2) a very detailed mapping of the spatial distribution of gullied lands. Firstly, the gully head-cuts were inventoried for the entire region using platforms such as Google, Esri, and Bing, through the QuickMapService plugin within QGIS 3.16 software, with the assistance of Landsat and Sentinel satellite images. Secondly, the automatic mapping of gullies was carried out using a 5 m high-resolution Digital Elevation Model and the Topographic Openness module offered by SAGA GIS software through QGIS software. As a result, we identified 5868 gully head-cuts for the Covurlui Plateau, with an average density of 2.57 gully head-cuts per square kilometer. Additionally, the identified gullies occupy over 3570 hectares, representing 1.57% of the total area. Overall, the topographic openness index proves to be an efficient tool for the semi-automatic extraction of gullies from high-resolution digital elevation models.

**Keywords:** gully erosion; gully inventory; semi-automatic extraction; topographic openness

## 1. Introduction

One of the primary challenges confronting the agricultural sector in the Moldavian Plateau is the issue of soil loss due to gully erosion. This phenomenon has significant implications for the sustainability and productivity of the region's agricultural land, posing a major concern for stakeholders in the sector, and represents a major threat to life, property, and infrastructure [1–3]. Gully erosion is considered one of the leading causes of sediment yield resulting from water erosion and unsustainable land management [4] and is commonly associated with both on-site and off-site impacts [2,5,6], such as terrain fragmentation, reduced trafficability [5] negative impacts on the hydrological functioning of catchments [7] and decreasing water quality and availability [6].

Over the past two decades, multiple investigations have been carried out to develop methodologies for the automatic or semi-automatic identification and mapping of gullies. These studies have been conducted worldwide, and they have primarily relied on remote sensing data, high-resolution digital terrain models, and machine learning statistical models. To extract gullies and identify regions susceptible to gullying, satellite imagery such

as Landsat and Sentinel and indexes to determine vegetation and soil characteristics have been utilized [8–10]. In addition, Google Earth images have been utilized to identify and extract gullies through Object-Based Image Analysis [11], watershed delineation, and terrain skeleton information [12]. Furthermore, geomorphometric derivates, including terrain roughness, terrain curvature [13], normalized slope, normalized elevation [14], and bidirectional shading [15], have been utilized in methodologies to identify gullies and extract gully shoulders. Machine learning techniques have also been utilized in various methodologies to identify and extract gullies, including Support Vector Machine [16,17], Neural Network [17,18], Maximum likelihood classification, Minimum Distance [16], Tree Decision Hierarchical Classification [19], and Random Forest [17]. In recent years, the emergence of techniques for creating high-resolution digital terrain models has led researchers to employ Lidar-derived digital elevation models for gully identification and extraction [20–22]. The development of an efficient methodology for the semi-automatic extraction of gullies from a given area is very important. Manual digitization of gullies is a time-consuming process, whereas the use of a semi-automatic extraction would facilitate the analysis of a larger area with greater accuracy and efficiency. With the semi-automatic extraction, the location and the extent of gullies can be identified, facilitating easier monitoring over time. The semi-automatic extraction will help by assessing the proximity of gullies to human infrastructure, and the extracted data would be vital in identifying areas where natural resources are likely to be affected. The temporal patterns of gully formation and evolution could be better understood by using this type of methodology and could enable the implementation of appropriate management strategies for the affected areas. Topographic openness is a topographic attribute that serves as an alternative to relief shading [23]. It can be calculated using digital elevation models (DEMs) and quantifies the area that is visible to the sky when looking from a given point on the terrain. According to Florinsky [24], topographic openness is a two-field specific variable comprising the morphology of the terrain and a set of sightlines between a given point of the topographic surface and the surrounding points. Daxer [25] states that this attribute is independent of any light source being an effective tool for visualizing prominent surface concavities and convexities. Moreover, compared to most geomorphometric parameters, it is less affected by the high-resolution Digital Terrain Models' noisiness [26].

Topographic openness has two visual perspectives: positive openness and negative openness. Positive openness is calculated using the mean value of zenith angles, and negative openness is calculated using the mean value of nadir angles. The nadir is the point on the terrain that is most sheltered from the elements and is surrounded by terrain features that obstruct the view of the sky. The zenith and nadir angles are calculated using the radius specified, depending on the topographic features that will be highlighted. A small radial limit will be useful for mapping ridges and gullies [25]. Compared to relief shading, which uses a light source and the position of azimuth, topographic openness effectively identifies microtopographic features, sunken paths, and hollows without a light source and excludes general topographic information. Topographic openness can be calculated for each pixel in the LiDAR DEM and then thresholded to identify pixels with a high value of openness. Positive openness consisting of high values for convex forms can be considered candidate gully pixels and can be further analyzed to determine if they represent actual gullies [27]. On the contrary, negative openness emphasizes topographic features that are below the surface topography, such as burial mounds and tumuli [28].

Since the mid-20th century, a number of studies have been conducted to investigate the inventory and distribution of gullies and gully head-cuts in various regions worldwide. According to Vanmaercke et al. [5], only eight research studies on the inventory and distributions of gullies and gully head-cuts have been conducted at the regional, national, and sub-continental scale, with seven of them focused on Europe and one on Africa. These studies have been conducted in the following European regions and countries: Slovakia [29], Poland [30], the southeastern part of Poland [31], the northwestern part of France [32], the Middle Volga region [33], and Hungary [34].

In Romania, the inventory comprising over 9000 gullies conducted by Rădoane [35,36] is considered the most significant and detailed research of this type in the Moldavian Plateau from eastern Romania [37]. Other gully inventories conducted in Romania include those by Mihaiu et al. [38] and Mircea [39] for the Buzău Subcarpathians [40] and Jurchescu [41] for the Getic Subcarpathians and the Getic Plateau, respectively. Most of these studies have relied on the use of topographic maps and aerial photographs and, more recently, on remote sensing data.

Against this background, our aim is to establish a methodology in order to achieve an (1) up-to-date, complete inventory of gully head-cuts and (2) a very detailed map of the spatial distribution of degraded lands by gully incisions based on advanced GIS techniques. Where possible, based on different time series input data, establishing the rhythm of the linear gully head retreat is another important issue taken into account

## 2. Study Area

The Covurlui Plateau constitutes the southernmost subunit of the Moldavian Plateau within eastern Romanian (Figure 1). Spanning an area of 2208 km², the plateau is bordered by the Prut Valley to the east, the Romanian Plain to the south, southwest, and west, the Bârlad Plateau to the northwest, and the Elan Hills to the northeast.

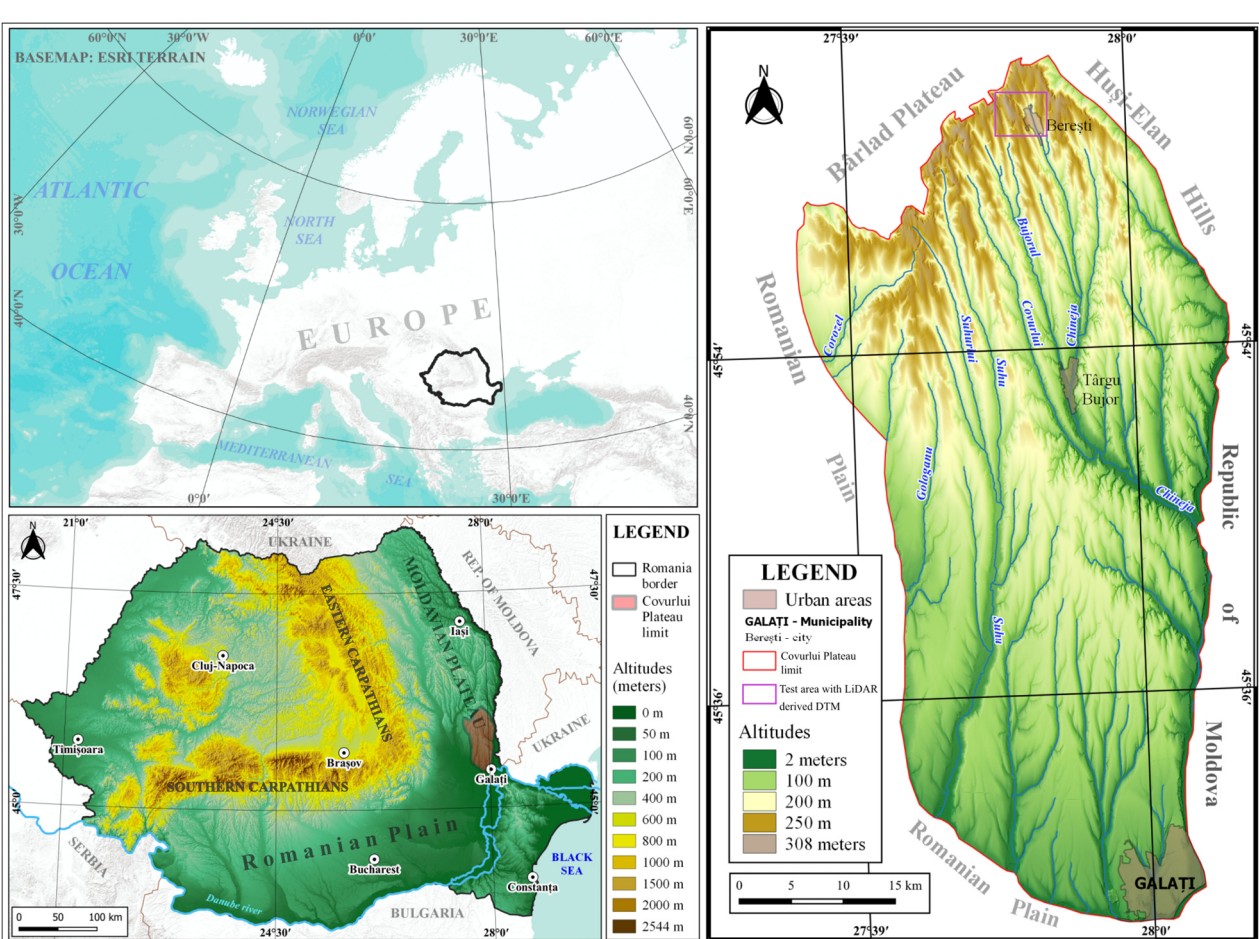

**Figure 1.** Location of the study area.

The lithology of the Covurlui Plateau is characterized by an alternation of sands with mudstones, loess-like deposits, and Quaternary gravels [42]. These deposits belong to the Pontian, Dacian, and Romanian layers, specific to the Middle and Upper Pliocene period (Figure 2). During the Pontian period, the region experienced a pronounced marine regression [43], which facilitated the deposition of a series of mudstone and compact sand

with a thickness of 50–70 m, over which Dacian deposits occur with the presence of red silts in a 2–10 m thick layer. Above the Dacian, a series of Romanian deposits consisting of a succession of yellow sands with mudstones, gravels, and sands having 30–50 m in thickness occur, being finally covered by a consistent layer of a maximum of 60 m of loess. As a result of the neotectonic uplift during the Romanian orogenetic phase that affected the whole Moldavian Plateau, the layers are slightly dipping from northwest to southeast. Thus, the oldest Pontian-Dacian and Romanian series are found on the slopes within the northern third of the region, while the younger Quaternary loess typifies the extensive interfluvial ridges, especially in the central and southern parts of the region [44,45].

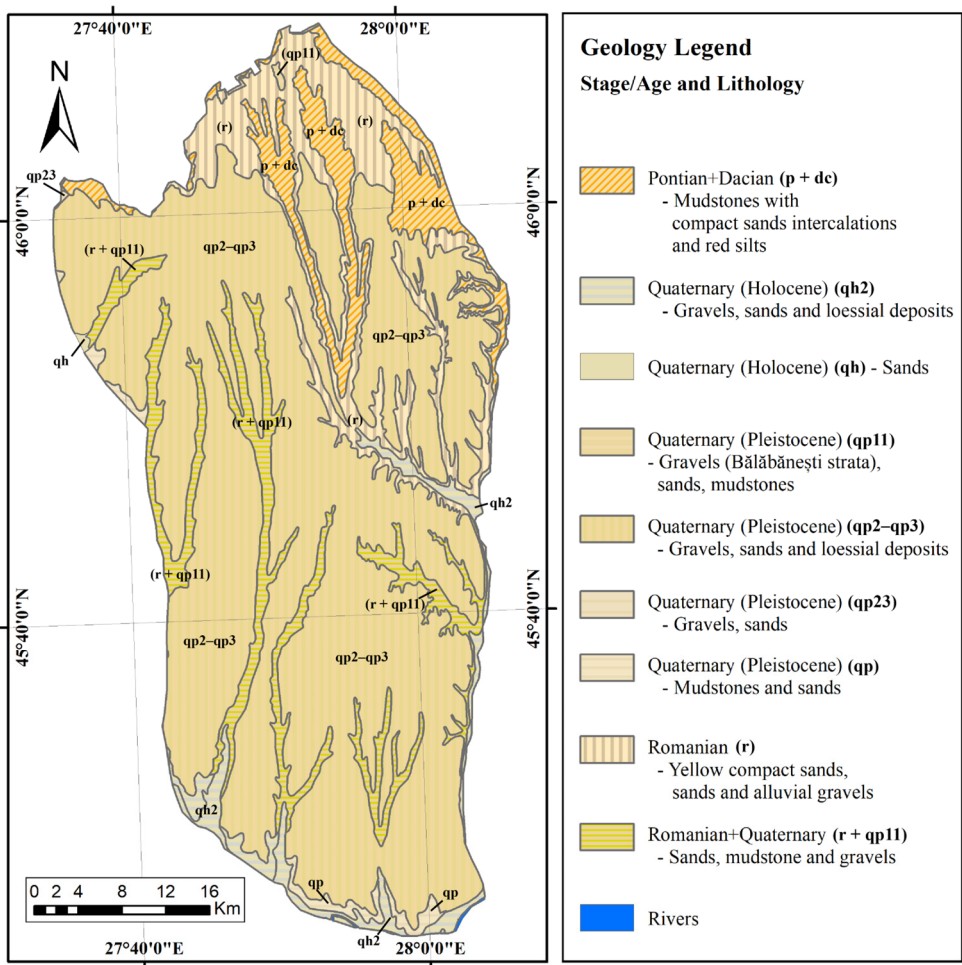

**Figure 2.** Geological map of the Covurlui Plateau (Geological Institute of Romania [46]).

In relation to the general monocline structure, the altitude gradually descends from a maximum of 308 m in the northern part of the region to a minimum of 2 m in the southeastern extremity. Consequently, from a geomorphological perspective, the northern half of the Covurlui Plateau exhibits a more developed hydrographic network which led to a higher degree of terrain fragmentation compared to the southern half, where extensive unfragmented flat surfaces prevail due to an underdeveloped hydrographic network. Correlated to this fact, the northern half of the plateau displays higher terrain slope values, ranging from 10 to 20%, in contrast to the southern half, which possesses lower slope values, usually fewer than 5%.

In the Moldavian Plateau, the average amount of annual precipitation varies between 450 and 700 mm/m$^2$/year. Over the entirety of the Covurlui Plateau, the amount of average annual precipitation varies between 450 mm in the central and southern low-altitude sectors and 500 mm in the northern, higher altitudes sectors. Although the overall amount of average annual precipitation is low, the distribution of precipitation varies greatly from day

to day and is unevenly distributed. An average drought period lasts for sixteen consecutive days, and there is an average of eight annual precipitation-free periods in the city of Galați, in the southern part of the Covurlui Plateau. In contrast to these situations, there are also days with extreme precipitation events of great intensity, where one-third or half of the monthly precipitation amount can be recorded in a few hours. The very large quantities of precipitation mainly come from torrential rains, which are one of the triggering factors for soil erosion in the Moldavian Plateau [47].

Consistent deforestation actions deployed during the 19th and early 20th centuries, as a result of the Treaty of Adrianopole, in 1829, and certain Agrarian Reforms, led to the conversion of forested lands into arable lands. Furthermore, the subsequent implementation of improper agricultural practices exacerbated widespread land degradation, particularly through soil erosion. In response, the Romanian government began implementing, starting in the 1960s, a series of measures aimed at mitigating accelerated erosion throughout the country, particularly within the Moldavian Plateau.

Nowadays, the Covurlui Plateau is one of the important agricultural areas in Romania [48]. According to the Land Cover dataset provided by the Copernicus Land Monitoring Service, over 83% of the total area of the region is occupied by agricultural fields, 57% being arable land [49] (Figure 3).

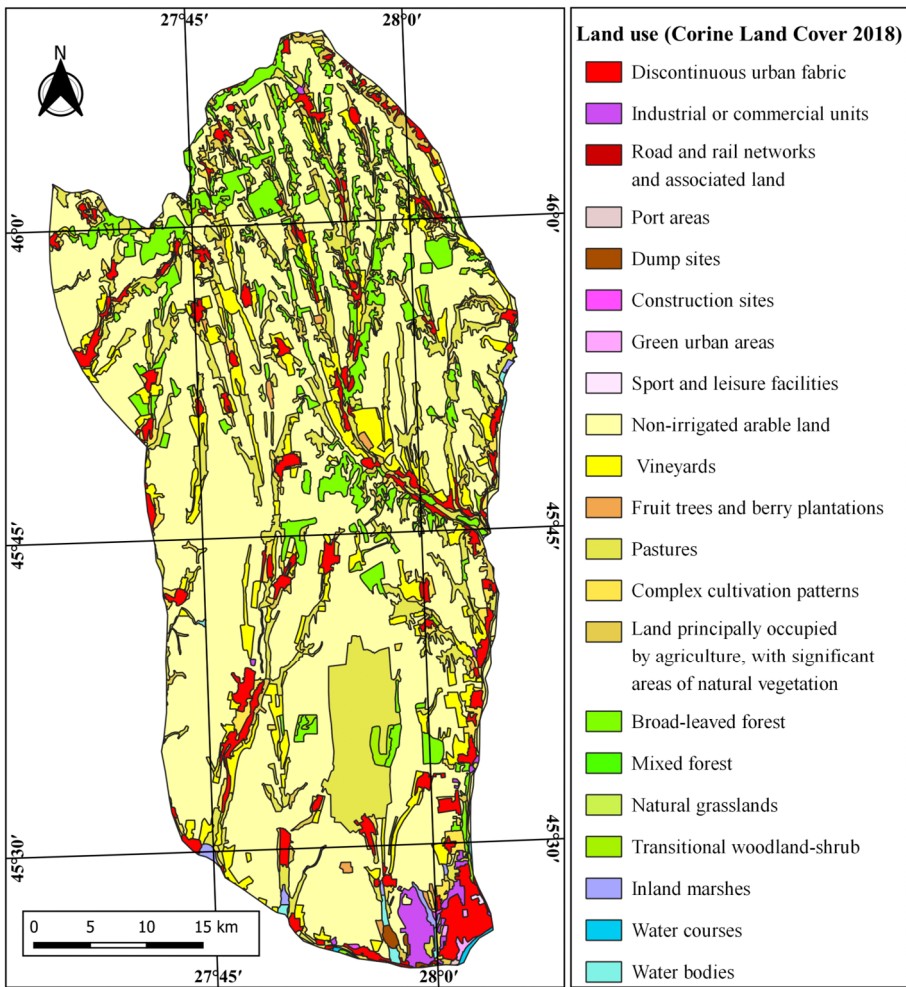

**Figure 3.** Land use map of the Covurlui Plateau [49].

Under this specific natural pattern, coupled with an intense and continuous improper human impact, land degradation has become, in the last two centuries, one of the most important environmental threats [39]. Due to its wide spread, intensity, and impact it has,

especially on agricultural land, gully erosion is the most important degradation process in the region [35,50].

Based on field observations and detailed measurements carried out over a period of over 40 years, Ionita et al. [51,52] established that the permanent gully systems within Moldavian Plateau consist of two main gully types, namely: (1) small, discontinuous, valley-sides gullies, which typify the northern part of the region and (2) large, continuous, valley bottoms gullies, which prevail in the southern area including the north half of Covurlui Plateau (Figure 4). Based on fourteen continuous gullies, they stated a mean linear gully head retrain of 7.7 m/yearover 60 years that was accompanied by a mean areal gully growth of 213 m$^2$/yr. Furthermore, the main controlling factors are depicted by (1) annual precipitation depth explaining 57% of linear gully retreat and 53% of areal gully growth rate and (2) contributing area with 33% and 43%, respectively.

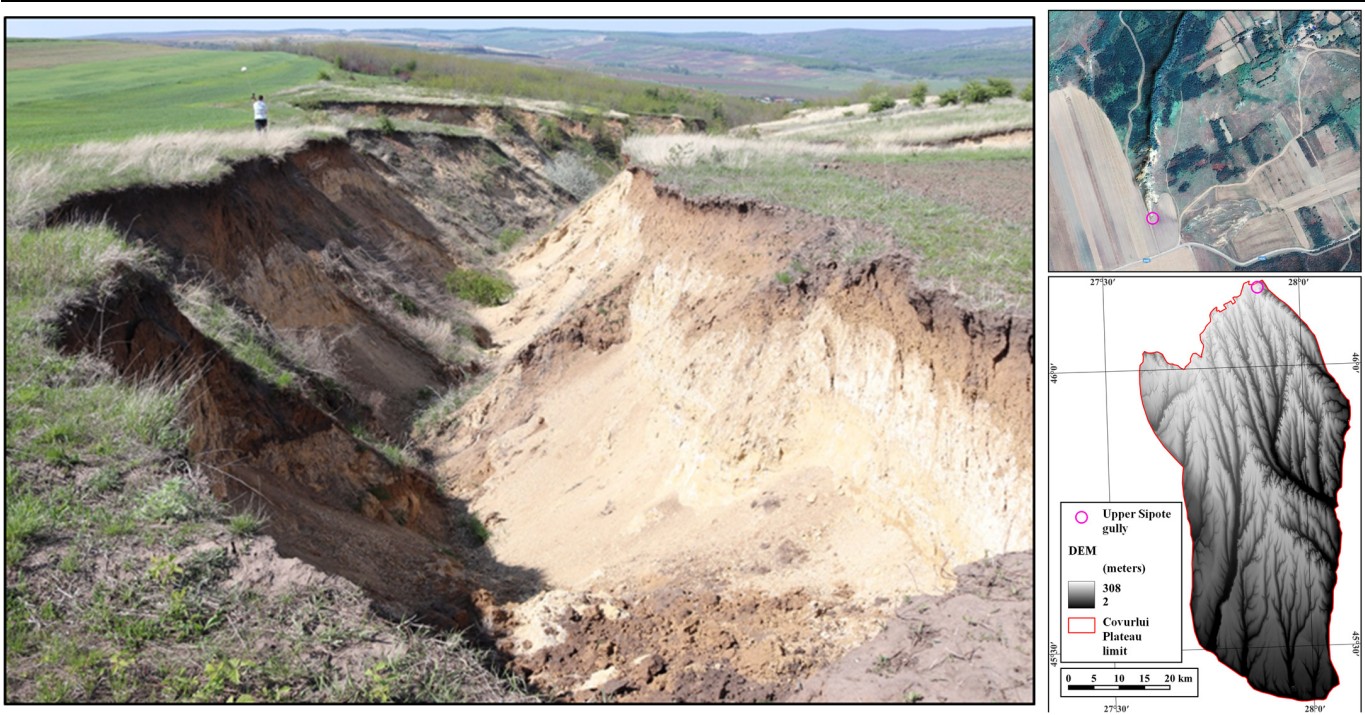

**Figure 4.** Upper Sipote gully from Horincea Catchment, Covurlui Plateau, developed since the end of the 19th century (3 May 2019).

## 3. Materials and Methods

In order to achieve the objectives of the research, two distinct methodologies were employed.

The first methodology was used to identify gully head-cuts, filter rills, and identify gully head-cuts that have retreated in the last decade. To accomplish this task, a variety of data sources were used, including Google Satellite imagery, Bing Satellite imagery, Esri Satellite imagery, aerial photos from 2005, 2012, and 2016, and Google Earth historical imagery (Figure 5). These data sources were analyzed using various software, including QGIS 3.16 with the Quick Map Services plugin, the National Agency for Cadastre and Land Registration (NACLR) geoportal (https://geoportal.ancpi.ro/accessed on 30 October 2022), and Google Earth Pro time series. These materials were used to create an inventory of gullies and rills. To filter the results and create a database of gully head-cuts, a 5-m resolution Digital Terrain Model provided by NACLR was used to analyze the depth and distinguish gullies from rills. After the inventory was filtered, Google Earth historical imagery was employed to create a database of gully head-cuts, identify gully head-cuts that have retreated in the last 10 years (between 2012 and 2022), and quantify the retreat

distance for each of them. In addition, supplementary validation for the largest gullies was performed on the most recent set of satellite images, such as Sentinel 2 or even Landsat. Despite the coarse spatial resolution (10 m for Sentinel 2 and 30 m for Landsat), the presence of a relevant number of large gullies could be validated via these spatial datasets.

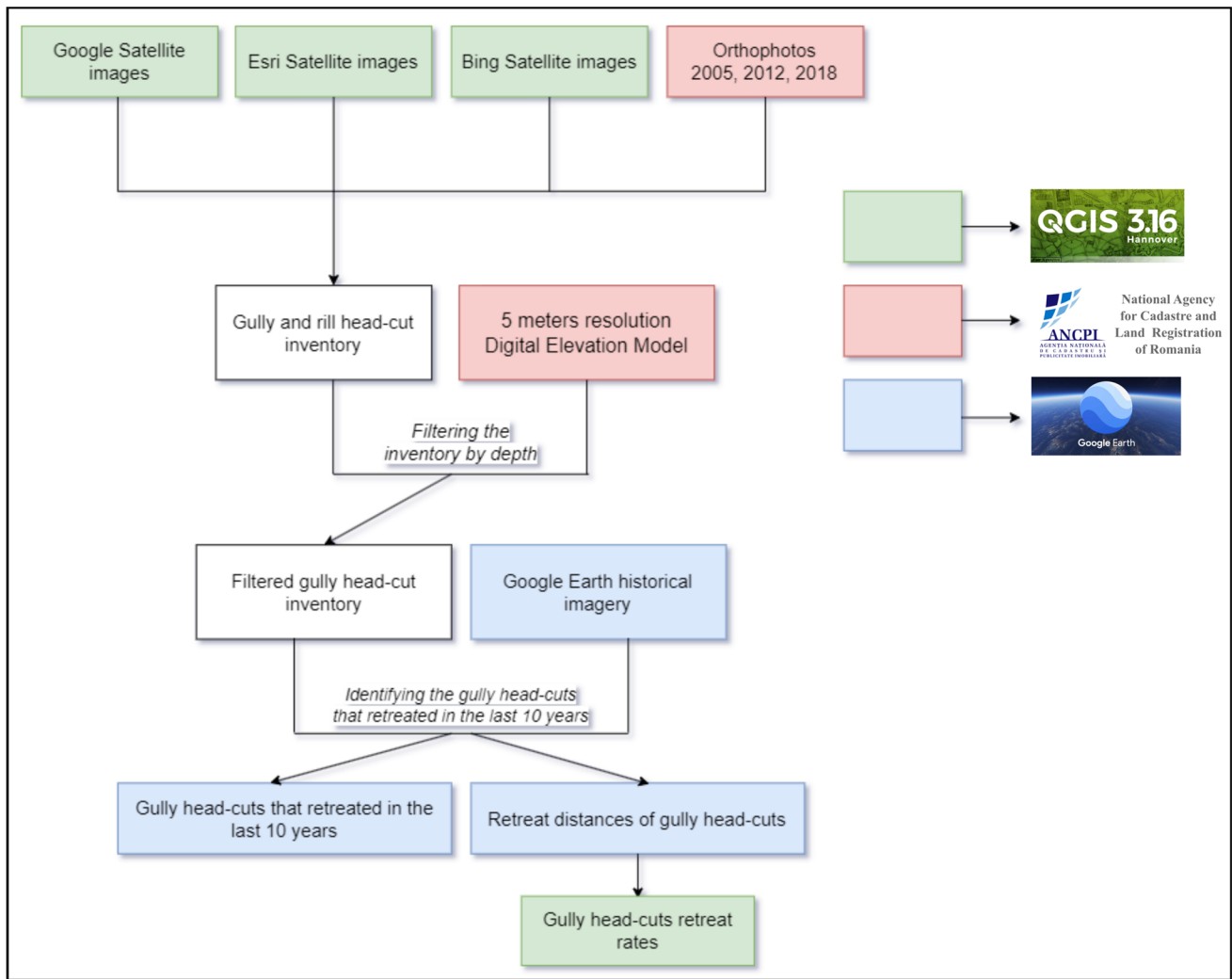

**Figure 5.** Methodological framework used to achieve the gully head-cuts inventory.

The second methodology that stands on a LiDAR-derived Digital Terrain Model (DTM) of 1 m resolution, representing a small test area within the northern region of the Covurlui Plateau, was used for the semi-automatic extraction of gully affected areas for the entire region (Figure 6).

First of all, the Topographic Openness component of the Terrain Analysis–Lighting and Visibility Module of SAGA GIS was used to create a Positive Openness grid. This was accomplished by setting a radial limit of 100 m and delineating a total of 8 aspect sectors (North, North-East, East, South-East, South, South-West, West, and North-West) using the Line Tracing method. Furthermore, the limit of the threshold necessary for gully extraction was identified at a value of 1.43, and the raster was subsequently exported from SDAT raster format to TIFF raster format. The limit of the threshold value was identified using the ability of the positive openness raster to extract gullies that are at least 1.5 m deep and filter out the rills that obviously have smaller depth values. The resulting TIFF raster was classified using QGIS's "Reclassify by Layer" tool inside the Raster analysis toolset by taking into account the previously identified thresholds for gully identification. The classified raster was converted into shapefile polygons using the QGIS Polygonize (raster

to vector) GDAL tool. The resulting polygons were filtered according to their individual area size to eliminate the redundant polygons resulting from pixel noise derived from the high-resolution DTM. For validation purposes, the polygons generated for the test area through the semi-automatic extraction process were compared to the polygons created by means of manual digitization.

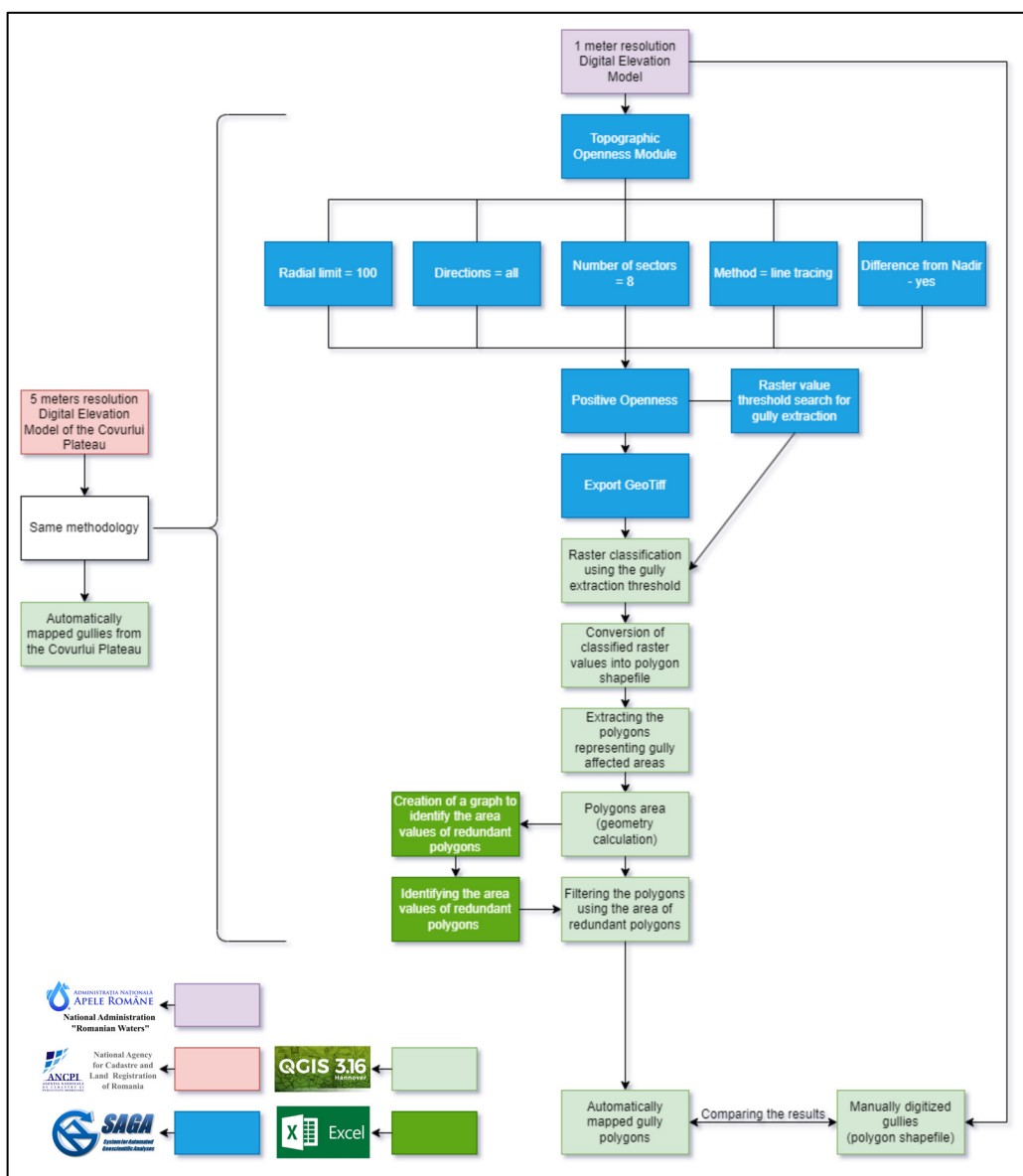

**Figure 6.** Methodological framework used for the semi-automatic mapping of gullied lands.

Finally, based on the results previously obtained in the test area, for the entire Covurlui Plateau, the 5 m resolution DTM delivered by the National Agency for Cadastre of Romania was used for the semi-automatic extraction of the areal distribution of gullies.

## 4. Results

*4.1. Results on Gully Head-Cuts Inventory*

The outcomes obtained by application of the first methodology consist of an inventory of 5868 gully head-cuts within the Covurlui Plateau (Figure 7). The areas with the highest densities of gully head-cuts were generally located in the northern half of the plateau, registering the absolute maximum value of 41 gully head-cuts per square kilometer (Figure 8).

Overall, the average density for the entire Covurlui Plateau was estimated at 2.57 gully head-cuts per square kilometer.

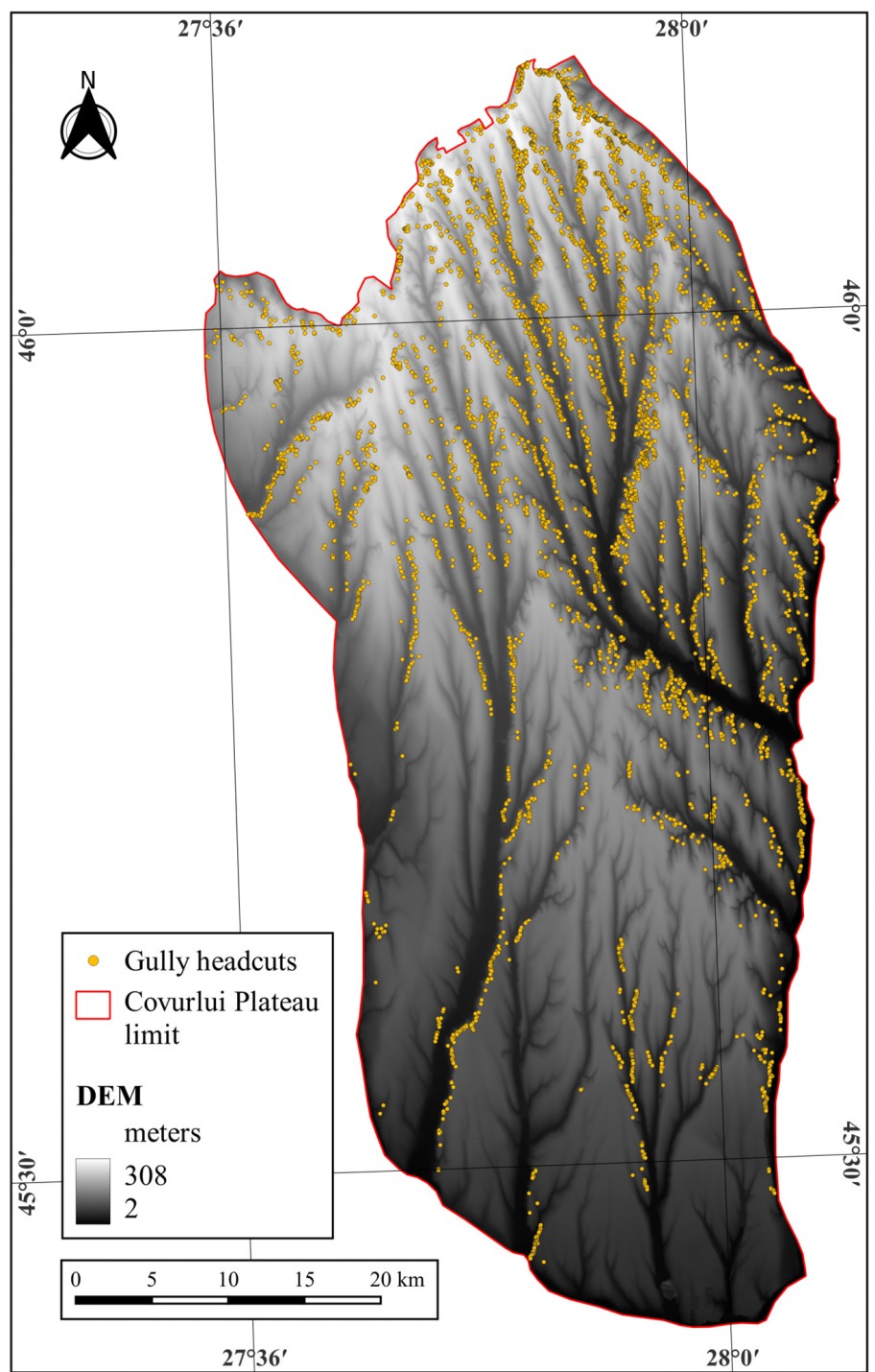

**Figure 7.** Map of gully head-cuts distribution in the Covurlui Plateau.

Based on Google Earth historical imagery in conjunction with the inventory of gully head-cuts, we selected 119 gully head-cuts that clearly indicate evolution over the past 10 years. Within limits given by the methodology and the quality of the input data, the retreat rates were subsequently calculated for these gully head-cuts.

The highest retreat rates were observed at two gully head-cuts with values ranging between 10.3 m/year and 12.72 m/year, respectively (Figure 9). Of the total number of gully head-cuts that retreated over the past 10 years, 38 (representing 32% of total) retreated

less than 1 m per year, 66 (55%) retreated between 1 and 3 m per year, 9 (8%) retreated between 3 and 5 m per year, 3 (3%) retreated between 5 and 7 m per year, one gully head-cut retreated between 7 and 10 m per year, while two gully head-cuts retreated more than 10 m per year.

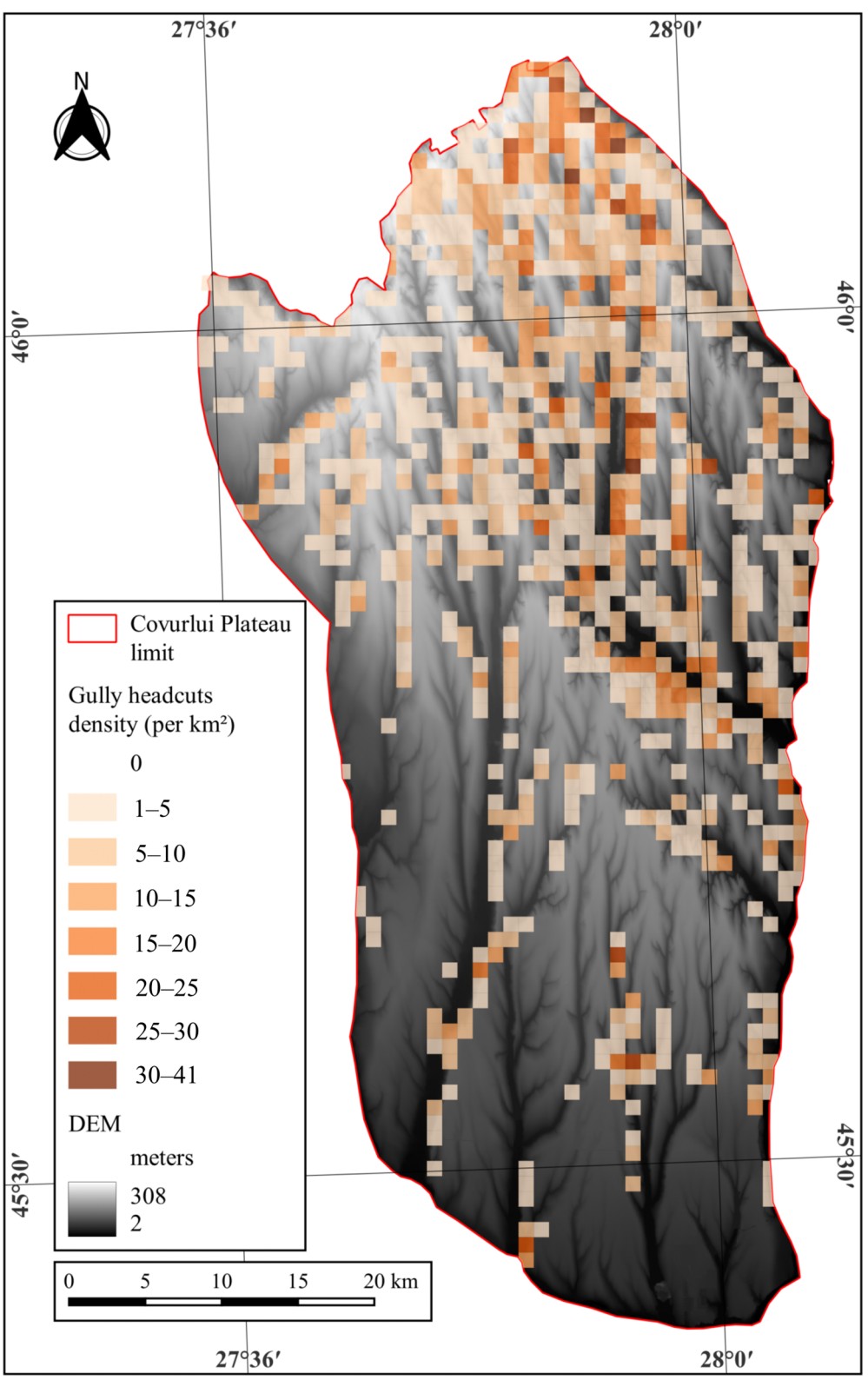

**Figure 8.** Map of the gully head-cuts density in the Covurlui Plateau.

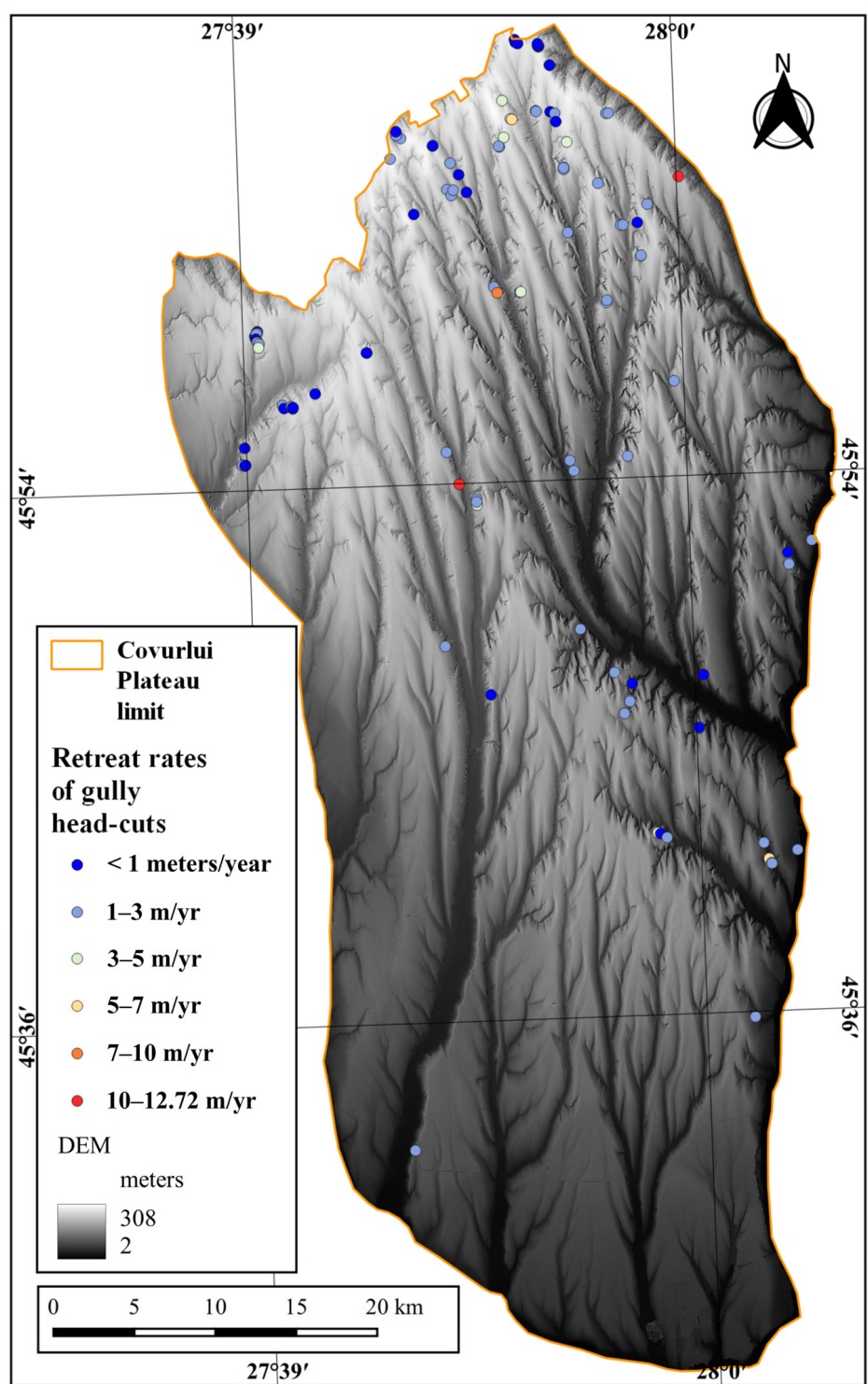

**Figure 9.** Map of the gully head-cuts retreat in the Covurlui Plateau.

An analysis of the land use pattern extracted from the Corine Land Cover 2018 dataset revealed that the active gully head-cuts were found especially on agricultural land (86%) and less on forest areas. Related to the main categories of agricultural areas, the majority of active gully head-cuts have developed on arable and arable primarily utilized for agricultural purposes categories (44%) and pastures (36%), respectively. Secondarily, only 8% of active gully head-cuts overlapped on vineyards (3%), complex cultivation patterns (2%), and orchards (1%).

### 4.2. Results on Extraction of Areal Gully Distribution

The second methodology yielded highly accurate results. The proposed methodology accurately extracted 95% of areas affected by gullying. Using the positive openness from the topographic openness module, only gullies with a minimum depth of 1.5 m were extracted. Practically, the difference between the steep slopes generated by the gully incision and the lower slopes of the surrounding undisturbed lands facilitated an accurate identification of the gully banks.

The total area occupied by gullies in the Covurlui Plateau was determined to be 3570 ha, representing 1.57% of the total area (Figure 10). By overlapping the obtained spatial data consisting of the up-to-date areal gully distribution with the 2018 Corine Land Cover dataset, we found that 37.7% of the gullied areas in the Covurlui Plateau are forested while 5.3% of them were located under shrub. The remaining affected areas were located under pastures (23%), arable land (18.5%), complex cultivation patterns (4.8%), and vineyards (4.7%).

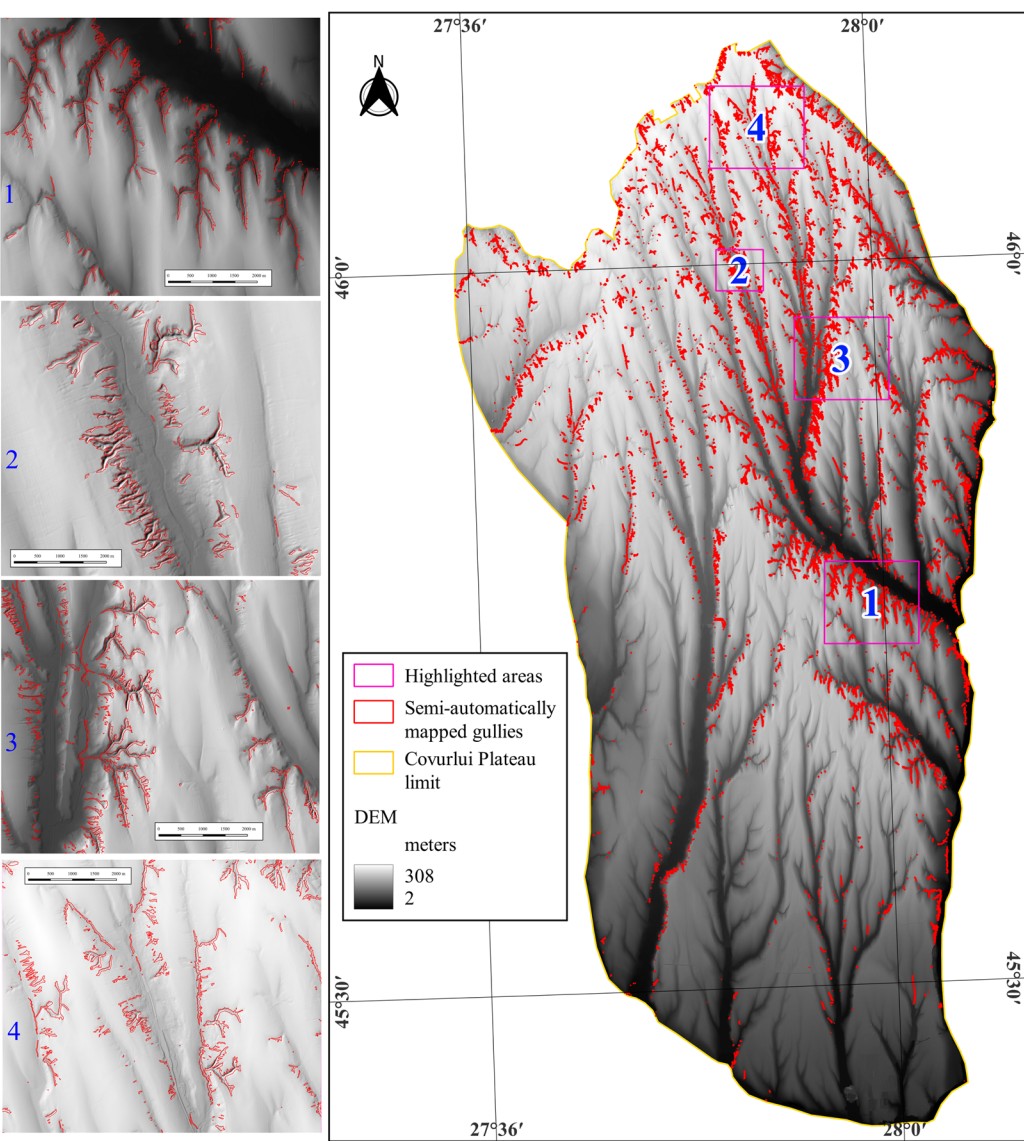

**Figure 10.** Map of areal gully distribution in the Covurlui Plateau.

## 5. Discussion

This method used to semi-automatically extract gullies in the Covurlui Plateau is a novelty for the Moldavian Plateau and Romania as well. The topographic Openness

method was used as an alternative to the shading method for identifying and classifying negative landforms. The use of high-resolution numerical models in large-scale gully inventory in Romania has not been achieved yet. The only important inventory so far is the one conducted by Radoane [35,36], which was made using topographic maps. Although an inventory of gullies in the Moldavian Plateau was conducted in the 1990s, it was based on topographic maps, aerial imagery, and field validation of a significantly small number of gullies. This study is a novelty because it was carried out using a digital elevation model with a spatial resolution of 5 m, which allows for the detailed analysis of the depths of negative landforms. This enables filtering out rills with depths of less than 1.5 m. Additionally, gullies in forested areas were mapped, which would be impossible with just the use of topographic maps and aerial imagery. The process is semi-automated, avoiding the subjective and time-consuming nature of manual digitization, and it has provided a significantly improved and updated overview of gully erosion in the Covurlui Plateau area. The topographic openness method helped us identify and extract gullies, especially in degraded areas of cuesta fronts, something that would have been impossible only by using slope values, shading, plan or profile curvature, roughness, or edge detection.

Slope gullies were best identified with this method (Figure 11, and valley bottom gullies were correctly identified up to areas with discontinuities resulting from silting. The gully head-cuts, which usually are the most active parts, were identified with very high accuracy, the semi-automatically obtained polygons being identical to the manually digitized polygons (Figure 12). Stabilized gullies, whose edges have been leveled by erosion and entered the valley stage, were not extracted with this method, which is an important advantage if we want very accurate results.

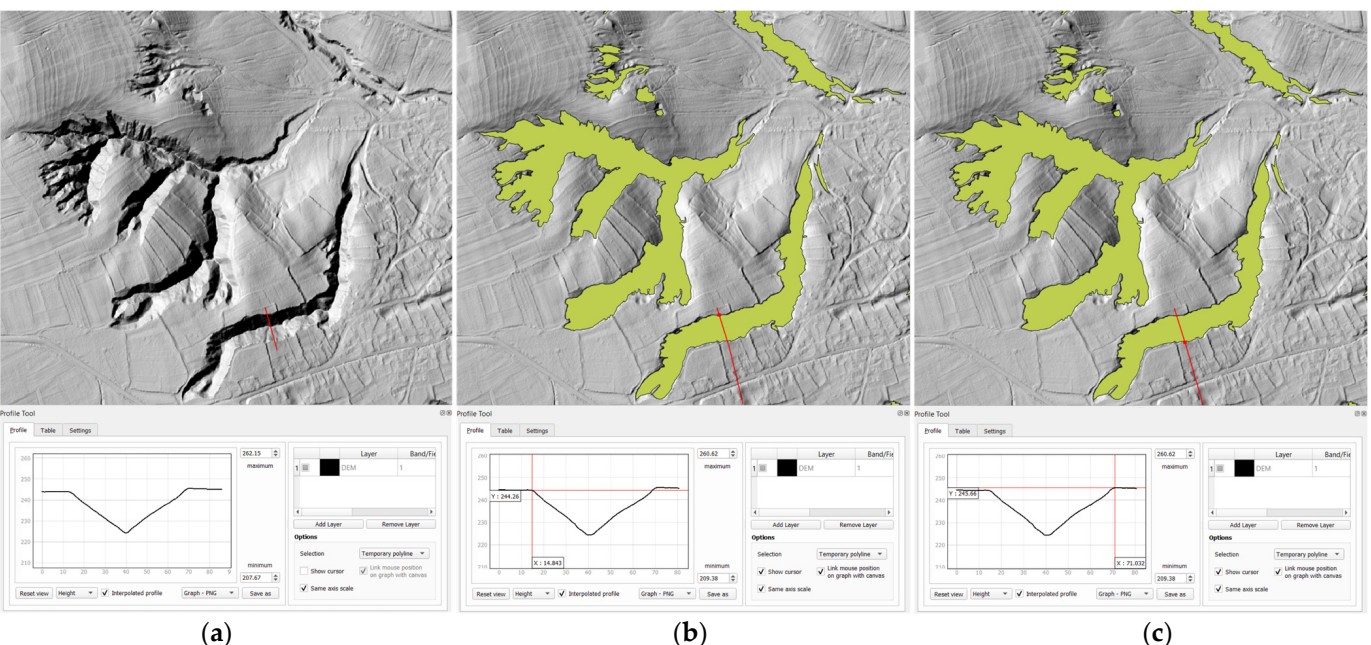

(**a**)                                                    (**b**)                                                    (**c**)

**Figure 11.** Topographic profile of gullies that were extracted using the topographic openness index and positive openness indicating: (**a**) hillshading of the area and semi-automatically extracted polygons of (**b**) left and (**c**) right gully bank.

Due to the predominant sandy-clay lithology and loess-like deposits in the Covurlui Plateau, the gullies in this area have very large widths and depths, also presenting very steep slopes between the edges and the bottom. Given these spatial characteristics of the gullies in the Covurlui Plateau, the 5-m resolution digital elevation model could be used in order to maintain accurate results. However, in areas with small gullies, the usage of the current methodology to numerical models with a 5-m resolution will give unsatisfactory results. For the best results, the geomorphometric characteristics of the gullies in a certain

area should be analyzed so that the level of detail of the numerical models used can be determined. This extraction method is semi-automatic because the results obtained with the Topographic Openness module must be filtered through the threshold method to eliminate redundant polygons and through centerlines, respectively shapefiles with roads, to filter ditches, irrigation channels, and excavations that could be identified as gullies through this method.

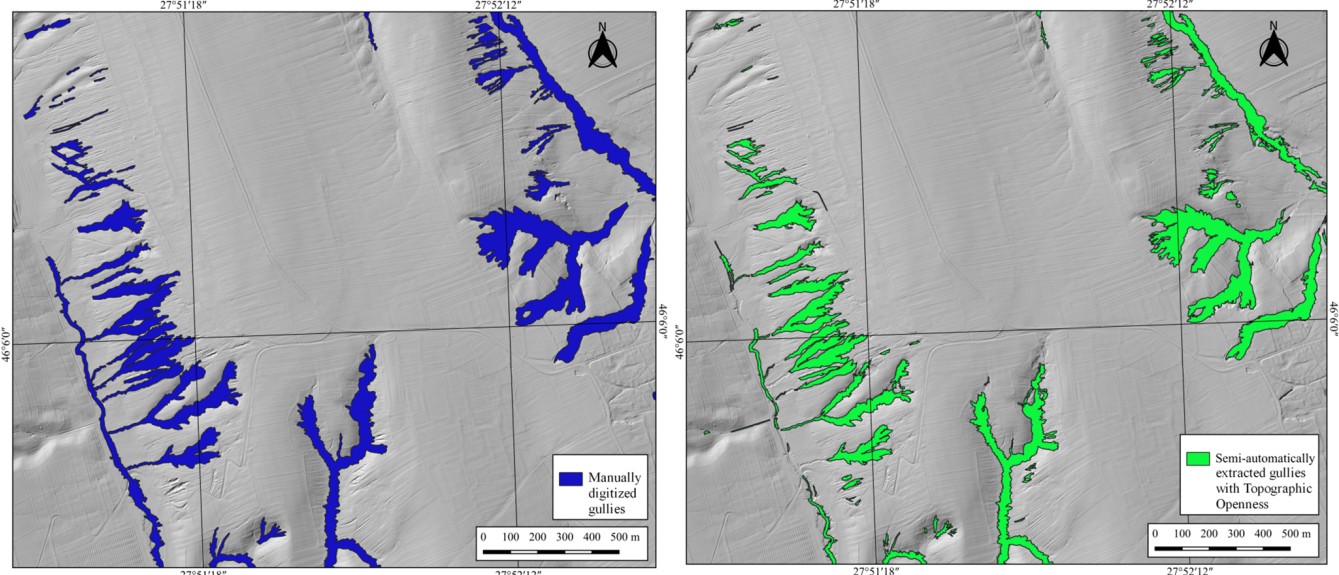

**Figure 12.** Gullies extracted using manual digitization (**left**), and semi-automatically extracted gullies using Topographic Openness (**right**).

After manually digitizing the gullies by using the digital terrain model derived from LiDAR point clouds for the test area in the northern part of the Covurlui Plateau, it was found that the gully-affected areas have a total cumulative area of 1,017,970 square meters. By applying the proposed methodology for gully extraction, it was determined that the gully-affected areas have a total area of 819,955 square meters. Therefore, by applying this methodology, 81% of the gully-affected area from the test zone was extracted. The surfaces that were not extracted using the proposed methodology include those of landslides or crumbling banks that affected the edges of the gullies, as well as those that do not have a depth greater than 1.5 m, which is the threshold that was used to differentiate gullies from rills. Additionally, the manual digitization of gully edges was not performed fully accurately due to limitations imposed by the resolution of the model and the hillshade raster. Based on these results, it can be stated that the proposed methodology has both limitations and advantages. The limitations include the inability to include gully crumbling banks and landslides within the gully boundaries and the inability to include gully head-cuts with depths less than 1.5 m within the gully boundaries. The advantages include the fact that this method extracts only negative landforms with depths greater than 1.5 m, excluding rills from the gully category, and the identification of gullies that could not have been mapped through manual digitization.

In order to compare the manually digitized polygons with those generated via semi-automatic means, through the "positive openness" side of the "topographic openness" function, four sample zones from the test area were used (Figure 13). For two of these areas, the ability of this methodology to extract side-valley gullies was tested; another one was used to test its ability to extract valley-bottom gullies and filter out rills; and the last one was used to test its ability to extract gullies with slopes affected by bank slumping or landslides.

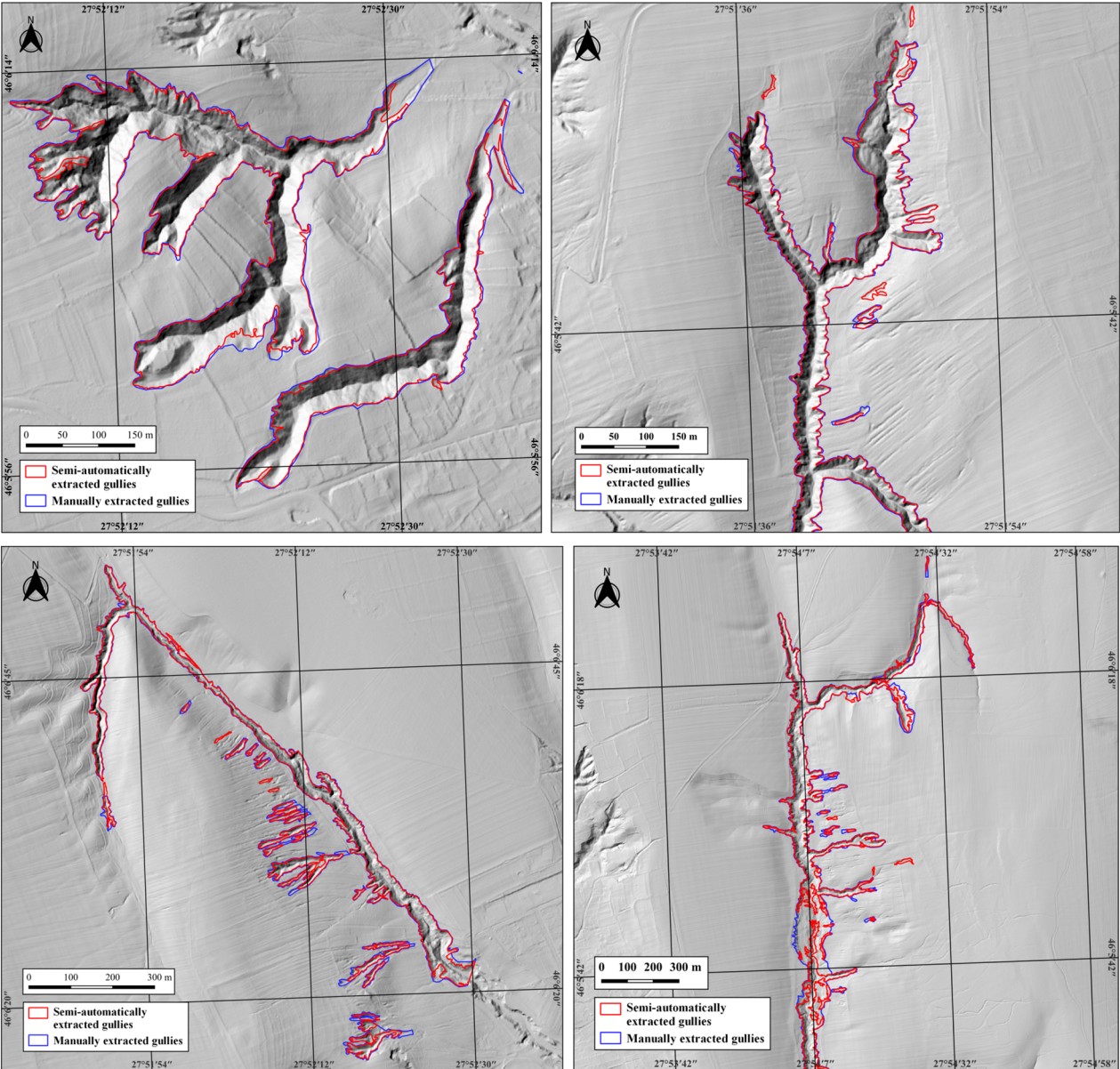

**Figure 13.** The four sample zones from the test area that were used for comparison between semi-automatically extracted gullies and manually extracted gullies (first area—**upper left**, the second area—**upper right**, the third area—**bottom left**, the fourth area—**bottom right**).

Inside the first area, the proposed methodology identified 88% of the total gully surfaces; the edges were correctly identified, especially in the area of the gully head-cuts. The only cases where the method did not correctly identify a gully head-cut were in areas where the gully has a low slope angle and the depth is lower than 1.5 m. In the second case, also applied to side-valley gullies, the methodological approach managed to identify 92% of existing gullies. The edges were also correctly extracted, and the only issues were related to gullies which are no deeper than 1.5 m. In the third area (including valley-bottom gullies and side-valley gullies), a total of 87% of the total gully area was extracted. Similar to the other areas, the specific gully parts that are not deeper than 1.5 m were not extracted, despite being part of an existing gully. Otherwise, the valley-bottom gullies were identified correctly, except for the head-cuts shallower than 1.5 m. Lastly, in the fourth area, the methodology applied extracted 70% of the gully areas, provided that the gully banks are more frequently subjected to slumping and landslides. The affected areas were not totally correctly mapped due to the fact that the nadir point could not identify the newly formed

bank after the occurrence of slumping and landslide processes, only identifying the bottom of the gully, which remained untouched. The body of the gully and the head-cuts were unaffected by slumping, and landslides were correctly extracted.

The majority of modern research from the international literature regarding the semi-automated extraction of gullies is based on using a mixture of satellite imagery and machine learning. Although most of this category of research provides significantly relevant and important results in the field, the greatest disadvantage is given by the incapacity to verify if the generated forms of linear erosion can be classified as "gullies". Despite being viewed as gullies from satellite imagery in the optical spectrum, in fact, they can be classified as rills with low depth values or small river valleys. In the current study, the methodological approach has identified only gullies in the Covurlui Plateau, while river valleys and rills with depth values lower than 1.5 m were not extracted.

In Covurlui Plateau, out of all the gully head-cuts that were identified, 70% are located on non-forested agricultural fields, and 30% are in forested areas. Taking into consideration the fact that the total surface of the agricultural lands is 1909 km$^2$, while the cumulated surface of the forests amounts to 197 km$^2$, the density value of gully head-cuts per km$^2$, is 2.1 on the non-forested areas, while the forested lands reveal a value of 8.8 gully head-cuts per km$^2$. The number of gully head-cuts that have retreated on the non-forested areas amount to a total of 106 (87% of the total head-cuts), while the head-cuts that have retreated on forested areas are 16, in total (therefore 13%). Therefore, the proportions have changed significantly, excluding the hypothesis that the number of active gullies depends on the overall area proportions of the different land-use categories.

Following the identification of gully head-cuts that have retreated in the last decade, an analysis was conducted to examine the impact of the catchment area, slope angle, slope length, and NDVI values on the retreat distance values of the gully head-cuts. In many studies on the susceptibility to gully erosion, they were identified as important controlling factors in the initiation and development of gully erosion [53–56]. However, research conducted by Ioniță et al. [40] on the influence of catchment area on gully head-cut retreat values in the Moldavian Plateau over a period of 60 years revealed that this factor does not appear to have a significant impact. Similarly, based on our data, a scatter-plot analysis of the values of catchment area and the values of retreat distances demonstrates that there is no correlation between these two variables in the Covurlui Plateau, as evidenced by the close proximity of the R-value to zero, indicating a very weak or non-existent statistical correlation (Figure 11). Moreover, three other scatter plots show very weak correlations between the slope angle, slope length and NDVI values, and the retreat distance of the active gully head-cuts (Figures 14–17). In addition, *p*-value coefficients were calculated, revealing values of 0.468 for catchment area correlation, 0.428 for NDVI correlation, 0.08 for slope length correlation, and 0.075 for slope value correlation. These values emphasize the lack of correlation between gully head-cut retreat values and each aforementioned parameter.

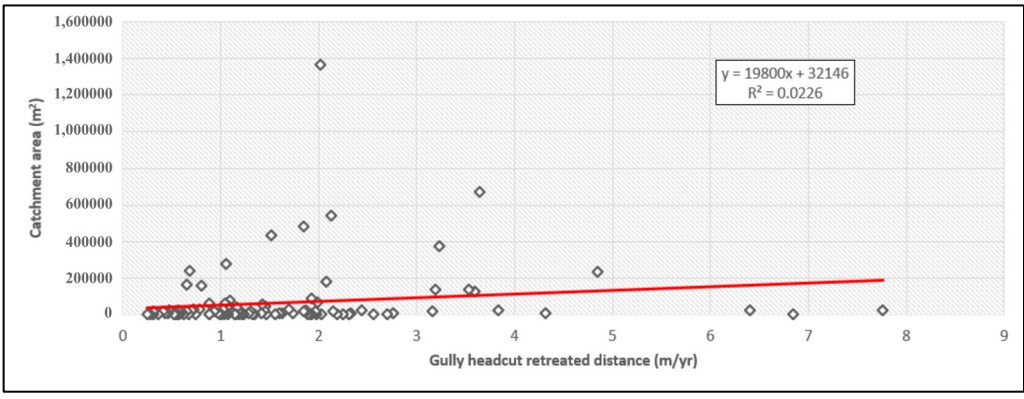

**Figure 14.** Plot of Catchment area to gully head-cut retreated distance per year.

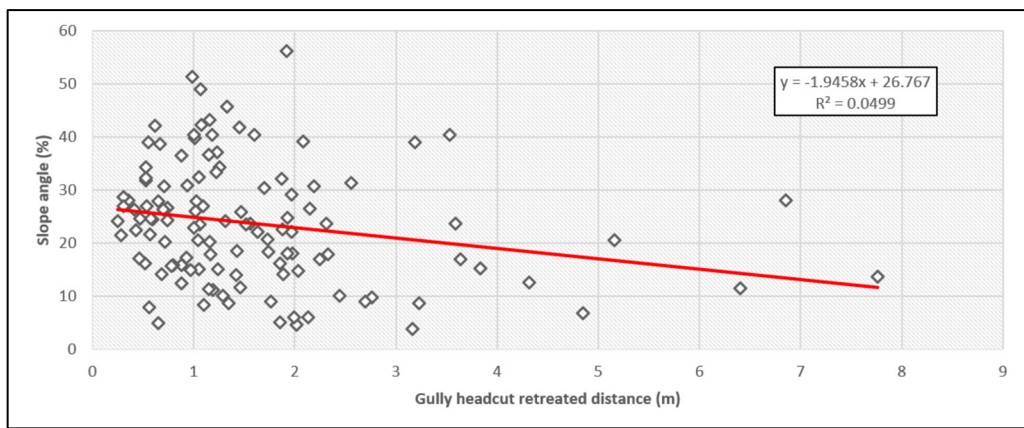

**Figure 15.** Plot of Slope angle values to gully head-cut retreated distance per year.

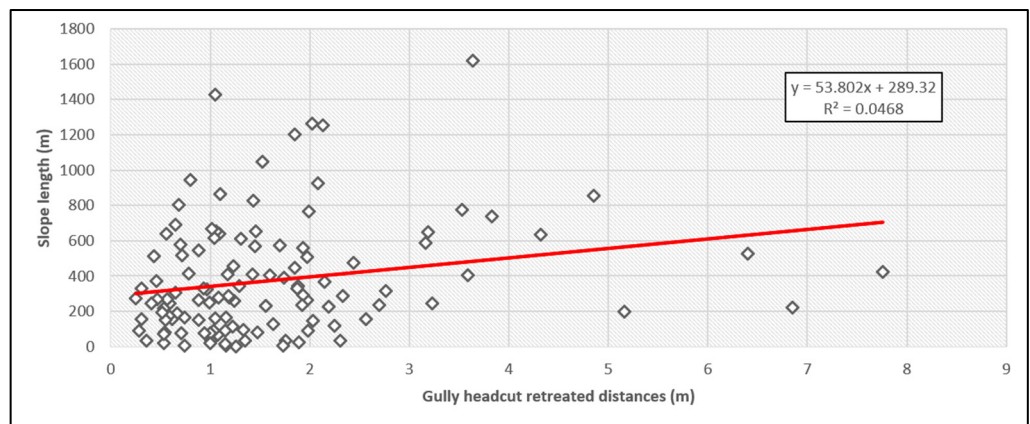

**Figure 16.** Plot of Slope length values to gully head-cut retreated distance per year.

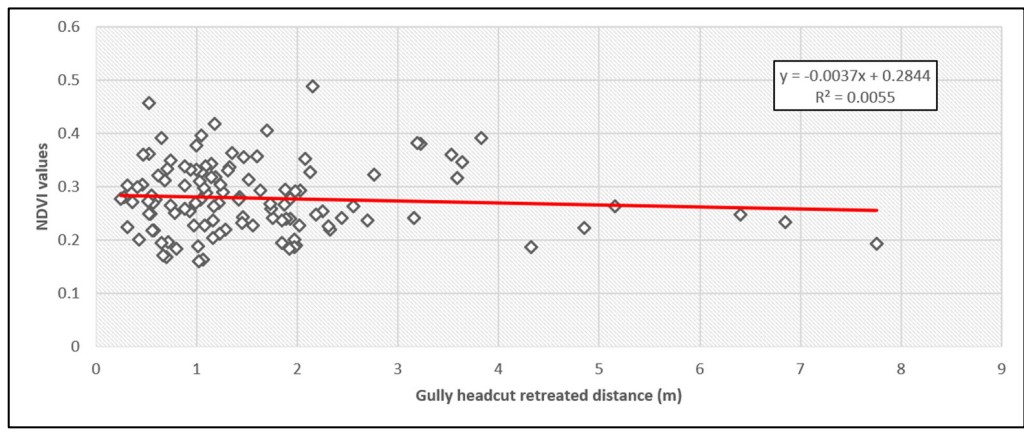

**Figure 17.** Plot of NDVI values to gully head-cut retreated distance per year.

As a possible cause, Ioniță et al. [40] indicate that the freeze-thaw phenomenon could be taken into consideration as the most important preparatory factor for gully head-cut retreat in the southern part of the Moldavian Plateau.

## 6. Conclusions

A gully inventory achievement is important from both theoretical and practical points of view. It provides information on the location, size, and characteristics of gullies, which are a type of landform created by erosion. This information can be used to understand the gullies' evolution, as well as to predict how they may change in the future. In addition, a

gully inventory can be used to identify areas that are particularly susceptible to erosion, which can inform management and conservation efforts. For society, gully inventory is important because gullying can cause a variety of negative impacts, including soil erosion and floodplain sedimentation that could generate important damage to the natural environment and human society.

The gully head-cut inventory is very useful for seeing the degree of fragmentation by gully erosion of a certain area and identifying the most active part of a gully. The inventory helps to determine the location, size, and characteristics of gully head-cuts and to monitor changes in their extent over time. This information can be used to calculate the rates of gully head-cut retreat, which is an indicator of the severity of gully erosion. Additionally, the gully head-cut inventory can provide important data for planning and implementing effective erosion control and rehabilitation measures, which can help to reduce the impacts of gully erosion on the environment and society.

The use of the Topographic Openness index, specifically the positive openness, represents an effective method for identifying and extracting areas impacted by gully erosion based on high-resolution digital elevation models. The identification and extraction are semi-automatic because it involves manual adjustment of parameters such as radius and threshold, as well as manual filtering of the extracted polygons. The positive openness index proves to be useful in detecting gullies and extracting the affected areas without the need for manual digitization, which can be a time-consuming process. These results are very useful in the calculation of the total area affected by gully erosion and the analysis of the evolution of individual gullies. The integration of high-resolution digital elevation models, generated using structure from motion techniques, satellite imagery, aerial imagery, and terrain measurements, in conjunction with the results obtained from this research, provides a robust step forward in research regarding gullying.

**Author Contributions:** Conceptualization, I.-C.C. and L.N.; methodology, I.-C.C., L.N. and L.B.-i.; software, I.-C.C., A.E. and L.B.-i.; validation, I.-C.C. and L.N.; formal analysis, I.-C.C., L.N., A.E. and L.B.-i.; investigation, I.-C.C. and L.N.; resources, I.-C.C. and L.N.; data curation, I.-C.C., L.N., A.E. and L.B.-i.; writing—original draft preparation, I.-C.C.; writing—review and editing, I.-C.C. and L.N.; visualization, I.-C.C., L.N. and A.E.; supervision, L.N.; project administration, I.-C.C. and L.N.; funding acquisition, I.-C.C. and L.N. All authors have read and agreed to the published version of the manuscript.

**Funding:** This research was funded by the Department of Geography, Faculty of Geography and Geology, "Alexandru Ioan Cuza" University of Iasi (Internal Research Program no. 1/2022).

**Data Availability Statement:** The data presented in this study are available on request from the corresponding author.

**Acknowledgments:** Acknowledgment is given by I.C. to the infrastructure support from the Operational Program Competitiveness 2014–2020, Axis 1, under POC/448/1/1 Research infrastructure projects for public R&D institutions/Sections F 2018, through the Research Center with Integrated Techniques for Atmospheric Aerosol Investigations in Romania (RECENT AIR) project, under grant agreement MySMIS no. 127324. This research was funded by a grant of the "Alexandru Ioan Cuza" University of Iasi, within the Research Grants program, Grant UAIC, code GI-UAIC-2021-02.

**Conflicts of Interest:** The authors declare no conflict of interest.

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
