# Peer review of "Gully Head-Cuts Inventory and Semi-Automatic Gully Extraction Using LiDAR and Topographic Openness—Case Study: Covurlui Plateau, Eastern Romania"

_land, doi:10.3390/land12061199_

Round 1

Reviewer 1 Report

The paper titled “Gully head-cuts inventory and semi-automatically gully extraction using LiDAR and Topographic Openness. Case study: Covurlui Plateau, eastern Romania” proposed a semi-automatic method for delineating gully areas by using high-resolution DEM. The experiments indicated that terrain openness is an ideal index for gully extraction. Generally, this topic is perfect for the scope of Land. The experiment design is appropriate, and the manuscript is well-organized. Therefore, I recommend accepting it after a minor revision.

I recommend the authors reorganize the Introduction section as it fails to capture the reader’s attention and give a clear statement of the main point of this manuscript. First, the harm of gully erosion and the significance of developing a gully extraction method should be emphasized at the beginning of the introduction section. Secondly, please don’t simply list the main work of each cited literature. You should classify the current studies based on the adopted methods or different data sources. Finally, I recommend citing more related studies which are not listed in references.

Line 108, km2, not km2

Figure 1. Please show the test area where the LiDAR-based DEMs were generated.

Figure 8. You can generate the Kernel density map (In ArcGIS), which could be a smoother surface.

Discussion. Previous studies investigated the DEM-based gully extraction using land surface parameters such as slope, catchment area, and hill-shading map. Whether the proposed method can perform better if more parameters are adopted deserves more discussion.

Recommended references

Castillo, C., & Gómez, J. A. (2016). A century of gully erosion research: Urgency, complexity and study approaches. Earth-Science Reviews, 160, 300-319.

Na, J., Yang, X., Dai, W., Li, M., Xiong, L., Zhu, R., & Tang, G. (2018). Bidirectional DEM relief shading method for extraction of gully shoulder line in loess tableland area. Physical Geography, 39(4), 368-386.

Liu, K., Ding, H., Tang, G., Song, C., Liu, Y., Jiang, L., ... & Ma, R. (2018). Large-scale mapping of gully-affected areas: An approach integrating Google Earth images and terrain skeleton information. Geomorphology, 314, 13-26.

Generally good, minor editing of English language required.

Author Response

Thank you for your comments. This is the cover letter for the revisions.

Reviewer 2 Report

The recent article entitled “Gully head-cuts inventory and semi-automatically gully extraction using LiDAR and Topographic Openness. Case study: Covurlui Plateau, eastern Romania” which deals with the extraction of gullies and its involved areas is an interesting topic in the discussion of semi-automatic extraction of gullies. In this article, the two methodologies and high-resolution data, including 1 meter and 5 meters, have been used, which has added to the scientific quality of the article. However, several items need to be revised and rewritten, which will certainly improve the quality of the article. Therefore, a major revision is recommended.

1.     Although the abstract has been quantitatively reported, the validation of the results still is unknown. Authors should control the results with validation metrics to justify the obtained results.

2.     The novelty of the study has not been well stated. The differences between the current study and other gully erosion mapping should be explicitly declared.  

3.     Results should be statistically compared and analyzed and just reporting the numbers and graphically comparison cannot be applicable. How authors can justify and trust the obtained results from the two methodologies?

4.      Discussion needs to deeply discuss and more comparison between the obtained results with other studies that have been reviewed in the introduction. There are some references in the literature that have been reported but none of them have been discussed and analyzed in the discussion. Moreover, state the strengths and weaknesses of the developed methodologies.  

5.     References should be updated based on the comments and also it is better to use the newest references such as 2023.   

Author Response

(The authors gave the same response as above.)

Reviewer 3 Report

I found the study worthy of discussion. Authors implemented several methods for obtaining gully inventory in a specific region. The main problem with the manuscript is related to the provided details and description of the methodology. Rather than naming the software, you must describe the application, library, algorithms, boundary conditions, and values used in the analysis. For instance, the Limit of the threshold for identifying gullies! is it classic 0.3 m or something else? Since your concern is to "establish a methodology" (Page 3, 2nd paragraph) you have to make it reproducible by the others. Otherwise, it would be more like a technical note provided for an institution that only cares about the results rather than methodology. There are also minor problems regarding the technical and grammatical contents of the text. e.g. giving an abbreviation before providing the full extension 

Minor grammatical and technical proofreading needed 

Author Response

(The authors gave the same response as above.)

Reviewer 4 Report

The article describes erosion mapping based on remote sensing data. The methods used, the territory under study and the results obtained are also interesting in the article. The authors studied the territory of the Kovurluy plateau. For this area, the problem of erosion is a topical research topic. The study consists of two parts. The first part is the search for gullies (throughout the Kovurlui plateau ) using satellite images and a digital elevation model with a resolution of 5 m/pixel. The second part is the search for gullies (on the test site ) using a digital terrain model with a resolution of 1 m/pixel, which was created on the basis of LiDAR surveys. Accordingly, the study has two main results. The first result is the inventory of gullies. The second result is a comparison of materials obtained from DEM and lidar data. The authors conclude that lidar data are more effective for mapping erosional forms.

The authors conclude that lidar data are more effective for mapping erosional forms. The data and methods of analysis used by other researchers are described. Much attention is paid to the publication of geographically close countries and regions. Based on this, the research problem is formulated and the tasks are clearly set. The introduction does not require any changes, clarifications or additions.

Section 2 describes the study area. The authors use the classical sequence of describing the territory under study. They pay attention primarily to the factors influencing the development of erosion. In addition to natural factors, anthropogenic factors in the development of erosion are described. Provides information about the history of land use and the current state of land use. The advantage of describing the study area is the availability of high-quality maps (Fig. 1, Fig. 2 and Fig. 3). The shortcoming of the description of the study area is the lack of information about the climate, primarily about precipitation. Although in line 171 the authors point to the influence of precipitation on the development of erosion.

Section 3 describes materials and methods. Methods are described completely and clearly. The authors took care of the reproducibility of the technique. They indicated all the software used, described the settings for data processing processes (including those for calculating topographic openness). The descriptions complement the graphic process diagrams (fig. 5 and fig. 6). This is a good approach to describing materials and methods. In section 3, one point remains unclear. The abstract mentions Landsat and Sentinel images. But in section 3 there is no mention of these satellite images.

Sections 4 and 5 present the results and discussion. Quantitative data on gullies and their growth are not in doubt. These data are of both applied and fundamental importance. But the article lacks a little statistical analysis of the data. So in Figures 11, 12, 13, 14 there is a coefficient of determination (R2). But there is no p-value. And without p-values, you can’t talk about statistical significance or insignificance (Lines 327-328).

Statistical analysis is needed for data on the number of actively growing (retreating ) gullies and their distribution by land use (Lines 247-253). Agricultural lands have more active ravines than forest lands. But maybe this is due only to the fact that agricultural land has a larger area than forest. It is necessary to compare the number of gullies with the proportion of land areas. To do this, you can do a chi-square test. An example of such a test is here: http://www.sthda.com/english/wiki/chi-square-goodness-of-fit-test-in-r (see "Answer to Q2 comparing observed to expected proportions").

I summarize everything written above, the following major changes need to be made to the article:

1) It is necessary to enter information about the climate in section 2, primarily about precipitation.

2) Section 3 explains the use of Landsat and Sentinel images.

3) A Chi-square test should be performed for the number of active gullies by land use and the area proportions of land use types. If the result is significant, then the distribution of ravines by land use is determined not by the area of land use, but by the erosion hazard of land use.

4) In figures 11, 12, 13, 14 you need to write the p-value (in addition to R2) or give this information in the text.

We also need minor changes in the text of the article:

1) The article needs to correct the numbering of the figures. Now, after Figure 10, there are Figures 15 and 16. And after Figure 16, there are Figures 11, 12, 13, 14. It is necessary to arrange the numbers of the figures in accordance with their appearance in the text. Don't forget to correct the references in the figure in the text.

2) In the caption to figure 4, it is desirable to add the coordinates of the place in the photo. I think readers will be interested in finding this place.

3) In Figure 9, you need to make a gradient height scale (as in Figures 8 and 10). The discrete scale with 2 classes does not correspond to how the relief is shown in Figure 9.

Author Response

(The authors gave the same response as above.)
